# Retrieval of the Complete Coding Sequence of the UK-Endemic Tatenale Orthohantavirus Reveals Extensive Strain Variation and Supports Its Classification as a Novel Species

**DOI:** 10.3390/v12040454

**Published:** 2020-04-17

**Authors:** Joseph G. Chappell, Theocharis Tsoleridis, Okechukwu Onianwa, Gabby Drake, Ian Ashpole, Phillipa Dobbs, William Edema, Frederick Kumi-Ansah, Malcolm Bennett, Rachael E. Tarlinton, Jonathan K. Ball, C. Patrick McClure

**Affiliations:** 1School of Life Sciences, University of Nottingham, Nottingham NG7 2UH, UK; joseph.chappell1@nottingham.ac.uk (J.G.C.); Patrick.Mcclure@nottingham.ac.uk (C.P.M.); 2Chester Zoo, Chester, Cheshire CH2 1EU, UK; 3Twycross Zoo, Atherstone, Warwickshire CV9 3PX, UK; 4School of Veterinary Science, University of Nottingham, Sutton Bonnington, Loughborough LE12 5RD, UK

**Keywords:** *Orthohantavirus*, hantavirus, high-throughput sequencing, virus discovery, field vole, United Kingdom

## Abstract

Orthohantaviruses are globally distributed viruses, associated with rodents and other small mammals. However, data on the circulation of orthohantaviruses within the UK, particularly the UK-endemic Tatenale virus, is sparse. In this study, 531 animals from five rodent species were collected from two locations in northern and central England and screened using a degenerate, pan- orthohantavirus RT-PCR assay. Tatenale virus was detected in a single field vole (*Microtus agrestis*) from central England and twelve field voles from northern England. Unbiased high-throughput sequencing of the central English strain resulted in the recovery of the complete coding sequence of a novel strain of Tatenale virus, whilst PCR-primer walking of the northern English strain recovered almost complete coding sequence of a previously identified strain. These findings represented the detection of a third lineage of Tatenale virus in the United Kingdom and extended the known geographic distribution of these viruses from northern to central England. Furthermore, the recovery of the complete coding sequence revealed that Tatenale virus was sufficiently related to the recently identified Traemersee virus, to meet the accepted criteria for classification as a single species of orthohantavirus.

## 1. Introduction

Orthohantaviruses are a large and diverse genus of viruses, belonging to the *Hantaviridae* family within the order *Bunyavirales*. The genome of orthohantaviruses consists of a linear, negative-sensed and single-stranded RNA, divided into three segments. The large (L) segment encodes a single RNA-dependent RNA polymerase, the medium (M) segment encodes a glycoprotein precursor and the small (S) segment encodes a nucleocapsid protein [1]. Historically, orthohantaviruses have predominantly been associated with rodent reservoir species [2]; however, they have increasingly been detected in other mammalian taxa, such as bats [3], shrews [4] and moles [5]. Each species of orthohantavirus is typically associated with a single reservoir species, where the infection is considered to be persistent and asymptomatic [6]. 

Several orthohantavirus species are capable of transmission into humans, through the inhalation of aerosolised contaminated excreta [7]. Human infection is thought to result in two forms of the disease, depending on the causative species; old-world species are associated with a primarily renal syndrome known as ‘haemorrhagic fever with renal syndrome’ (HFRS), whilst new-world species are associated with pulmonary disease, ‘hantavirus pulmonary syndrome’ (HPS) [8]. However, an overlap of clinical presentations between the two syndromes has led to suggestions that they should be reconsidered as a single clinical syndrome, hantavirus fever (HF) [9]. The severity of HF can vary significantly; Puumala virus (PUUV) infection, for example, causes a mild, often sub-clinical disease [10], whilst new-world species, such as Sin Nombre virus (SNV), has a case fatality rate of 35% [11]. There are four species known to cause HF in Europe; Seoul (SEOV), Dobrava-Belgrade (DOBV), Tula (TULV) and Puumala (PUUV) [12]; the reservoirs associated with these viruses are the brown rat (*Rattus norvegicus*), yellow-necked mouse (*Apodemus flavicollis*)/striped field mouse (*Apodemus agrarius*), common vole (*Microtus arvalis*) and the bank vole (*Myodes glareolus*), respectively. Except for the TULV-associated common vole, which is geographically restricted to the Orkney Islands in Scotland, each of the reservoir species associated with these viruses are present in the United Kingdom (UK). However, of these viruses, only SEOV has been detected in the UK [13]. 

HF has been reported sporadically throughout the UK, including England [14], Scotland [15] and Northern Ireland [16], though the causative species could not be confirmed due to cross-reactivity of the serological assays used to diagnose the orthohantavirus infections [17]. The first orthohantavirus linked to HF in the UK was in 2011 when a novel strain of SEOV was isolated from wild rats captured on the farm of a patient with suspected HF [18]; SEOV was then detected in pet rats belonging to a patient with serologically confirmed HF in 2013 [19]. Furthermore, a novel vole-associated hantavirus related to TULV and PUUV—Tatenale virus (TATV)—was identified in field voles (*Microtus agrestis*) captured in northwest England in 2013 [20] and again in northern England in 2017 [21]. However, fragments of less than 400 nucleotides were retrieved for two of the three genomic segments, meaning that phylogenetic analysis of this virus was limited. In 2019, an orthohantavirus was detected in German field voles—Traemersee virus (TRAV)—and was suggested to be a strain of Tatenale virus. However, the aforementioned paucity of published TATV sequence data has precluded any accurate comparison between TATV and TRAV [22]. To better understand the prevalence and phylogeny of Tatenale virus, we performed in-depth sampling and analysis of various rodents living in the UK.

## 2. Materials and Methods 

### 2.1. Samples 

Rodents were caught at two semirural sites in the UK: Leicestershire (Site 1, 52.6524° N, 1.5291° W) and Cheshire (Site 2, 53.2273° N, 2.8844° W). Seventy-two rats (*Rattus norvegicus*), 224 mice (*Mus musculus*) and 12 field voles (*Microtus agrestis*) were collected from Site 1 between May 2013 and October 2014. Eight rats, 119 field voles, 93 wood mice (*Apodemus flavicollis*) and 3 bank voles (*Myodes glareolus*) were collected from Site 2 between June 2013 and July 2016.

Rodents were captured as part of routine pest-management at both sites. Ethical approval for collection of rodent tissue had been previously been granted [23] by the University of Nottingham School of Veterinary Science Ethical Panel, reference numbers 1602 151102 and 1786 160518.

### 2.2. Nucleic Acid Preparation

Sections of lung and kidney tissue, approximately 1 mm^3^ were collected, and RNA was extracted using GenElute™ mammalian total RNA miniprep kit (Sigma Aldrich, St Louis, MO, USA), following the provided protocol. RNA was quantified using a NanoDrop spectrophotometer (ThermoFisher Scientific, Waltham, MA, USA). cDNA was synthesised from the RNA using RevertAid reverse transcriptase (ThermoFisher Scientific) following the provided protocol.

### 2.3. RT-PCR Screening

Two-step RT-PCR was performed on the samples, using a degenerate primer pair (HanSemiF: GAATATATATCNTAYGGDGGDGA and HanSemiR: CTGGTGACCAYTTNGTNGCAT) designed in-house to target a 178 bp region of the L segment of all known hantaviruses. PCR reactions contained 0.5 μL of cDNA, added to 1.25 μL 10× PCR buffer, 0.06 μL of HotStarTaq DNA polymerase (QIAGEN, Hilden, Germany), 0.5 μL 10 mM dNTP’s (Sigma Aldrich), 0.5 μL each of forward and reverse primer (10 Pmol/μL), and 9.19 μL of water for a total volume of 12.5 μL. Cycling conditions were 95 °C for 15 min, 55 cycles of 94 °C, 51 °C and 72 °C for 20 s each, followed by 72 °C for 10 min.

A second degenerate pan-hantavirus assay, targeting a different, larger region of the L segment, was used to confirm positive PCR results. Primers were sourced from Klempa et al. [24] (Han-L-F1: ATGTAYGTBAGTGCWGATGC and Han-L-R1: AACCADTCWGTYCCRTCATC); PCR reactions contained the same concentration of reagents and primers; cycling conditions were modified to 95 °C for 15 min, followed by 55 cycles of 94 °C (30 s), 53 °C (50 s) and 72 °C (30 s), finished with a final extension of 72 °C (10 min). This assay produced an amplicon of 452 bp.

### 2.4. High-Throughput Sequencing

The orthohantavirus positive field vole from Site 1 was selected for high-throughput sequencing (HTS). NEBNext^®^ rRNA depletion kit (Human/Mouse/Rat) (New England Biolabs, Ipswich, MA, USA) with RNA sample purification beads (New England Biolabs, Ipswich, MA, USA) was used to deplete host ribosomal RNA from the sample. Sequencing libraries were then created from the depleted RNA using NEBNext^®^ Ultra™ II directional RNA library prep kit for Illumina (New England Biolabs). Libraries were sent to SourceBioscience (Nottingham, UK) and sequenced with an Illumina HiSeq 4000. Each read length was 2 × 150 bp, and the insert size was 200 bp on average. All generated sequence data were analysed using the Geneious Prime 2019.0.4. Generated reads were mapped to reference orthohantavirus sequences downloaded from GenBank.

### 2.5. Retrieval of TATV’s Complete Coding sequence (CDS) Using PCR Primer-Walking

The complete coding sequences (CDS) of the Tatenale virus retrieved via HTS was used as a reference sequence to design primers to retrieve the CDS of the TATV strain from Site 2, as funding was not available for HTS of both samples.

### 2.6. Phylogenetic Analysis

Nucleotide sequences for each segment of both TATV strains were aligned with a full-length coding sequence for representative Arvicolinae-associated orthohantaviruses and a non-arvicolinae orthohantavirus (Andes Virus) outgroup, using the MUSCLE function in MEGAX [25]. MEGAX was then used to find the best-fit substitution model for each alignment of sequences, and the model with the lowest Bayesian information criterion scores was considered the most appropriate.

Maximum likelihood trees were created with a GTR+G+I model, using MEGAX software. Robustness was assessed using bootstrap resampling (1000 replicates).

The pairwise evolutionary distance (PED) values of TATV and related orthohantaviruses were calculated using a WAG amino acid substitution model, in the PhyML [26] plugin on Geneious Prime. These calculations were based on a concatenation of the complete coding regions of S and M segments from the same virus.

## 3. Results

### 3.1. Detection of Orthohantavirus RNA by RT-PCR

Orthohantavirus RNA was detected in a single field vole from Site 1 (8.3%) and twelve field voles from Site 2 (10%). No orthohantavirus was detected in any house mouse (*n* = 224), wood mouse (N = 93), brown rat (*n* = 80) or bank vole (*n* = 3) samples.

The sequenced HAN-L amplicons from the Site 2 voles were highly conserved; the two most dissimilar were 97.7% identical at the nucleotide level. Comparison of the HAN-L amplicons between the two sites were more divergent, with a nucleotide homology of 86.3% to 89.3% between the single Site 1 virus and Site 2 viruses. We named the strain from Site 1 ‘Norton-Juxta’ and the strain from Site 2 ‘Upton-Heath’, reflecting the geographic origins of the two strains.

BLASTn searches of the 452 bp HAN-L PCR amplicons showed a high level of similarity to Tatenale orthohantaviruses. The Norton-Juxta strain was 87.3% identical to ‘Tatenale virus strain B41′ and 86.5% identical to ‘Kielder hantavirus kld-1′ at the nucleotide level, whilst the Upton-Heath Strains were 94.9–96.9% identical to B41 and 84–86.6% identical to kld-1.

### 3.2. Recovery of Complete TATV CDS

A total of 62,191,960 reads were sequenced from an uHTS library created from the lung tissue of the Norton-Juxta positive field vole. A total of 27,279,217 reads remained following pair-merging and quality processing. Mapping of these reads to reference sequences for each segment resulted in a total of 94,706 reads, representing 2.5% of filtered reads. The complete coding sequence of each segment was recovered. The L segment was 6465 nucleotides in length (Genbank Accession number MK883761), whilst the M segment was 3447 nucleotides (MK883759), and the CDS of S was 1302 nucleotides (MK883757).

Complete CDS of the L (MK883760) and S (MK883756) segments of TATV Upton-Heath was recovered through PCR primer-walking, these sequences were the same length as those for TATV Norton-Juxta. Almost complete CDS of the M segment (MK883758) was recovered, missing 90 nucleotides from the 3′ end of the CDS.

### 3.3. Analysis of Complete TATV CDS

Comparison of the complete L and S segments and almost-complete M segments of the two strains revealed a nucleotide similarity of 90.6%, 94.1%, and 91.3%, respectively. Phylogenetic analysis of the three segments, with complete Arvicolinae-associated orthohantaviruses, showed that both Norton-Juxta and Upton-Heath TATV clustered closely with Traemersee virus, forming a distinct clade, and supported with strong bootstrap values in the L (Figure 1A), M (Figure 1B) and S segments (Figure 1C). Nucleotide and amino acid similarities between both TATV strains and closely related orthohantavirus species are shown in Table 1. Pairwise evolutionary distance (PED) analysis of the concatenated S and M segments of Norton-Juxta and other vole-borne orthohantaviruses showed values of between 0.12 and 0.27. The PED values between Norton-Juxta and TRAV were 0.05.

Comparison with the partial S sequence of TATV-B41 showed a nucleotide similarity of 98.7% with Upton-Heath and 93.9% with Norton-Juxta. Phylogenetic analysis with the partial L and S segments for other TATV strains showed that Norton-Juxta formed a novel lineage in the phylogenies of both segments, whilst Upton-Heath clustered closely with the previous B41 strain (Appendix A).

## 4. Discussion

This was the first reported recovery of complete coding sequences for TATV in the UK. Based on a genetic divergence, we proposed that this virus represented an additional strain of TATV, tentatively called Norton-Juxta, which extended the known range of TATV from northern to central England. The detection of diverse TATV in field voles, but not other species of rodents sampled from the same sites, strengthened evidence of field voles as the primary reservoir of the virus. The high similarity between the available sequence data for TATV B41 and the corresponding sequence from TATV Upton-Heath, together with the close geographic proximity of the collection sites, indicated that the two viruses might be co-circulating within the same population of field voles.

Orthohantavirus species’ demarcation criteria of >7% AA divergence across S and M segments [27], as well as stricter criteria of a PED lower than 0.1 in the concatenated S and M segments [28], have been suggested. As the PED values between the complete Norton-Juxta strain of TATV and TRAV was below the 0.1 speciation threshold, this confirmed that both TATV and TRAV were members of the same viral species, as was hypothesised by Jeske et al. [22]. Though TRAV was the first strain with complete CDS, TATV was detected several years prior and is more established in the literature. Thus, we proposed that the species in which TATV and TRAV belonged to be named Tatenale orthohantavirus.

There is serological evidence of human infection with PUUV- or SNV-like viruses in the UK [29], though there has been no molecular evidence of these viruses in either humans or rodents. A previous study has reported that blood from a vole infected with TATV B41 is cross-reactive with PUUV, which suggests that PUUV/SNV seropositive humans may have been the result of TATV infection [20]. Until now, the paucity of sequence data has precluded significant further investigation of TATV. Recovery of the complete coding sequence for each of the segments, particularly the glycoproteins encoded in the M segment, will allow for in vitro studies to further explore the zoonotic potential of the virus.

## Figures and Tables

**Figure 1 viruses-12-00454-f001:**
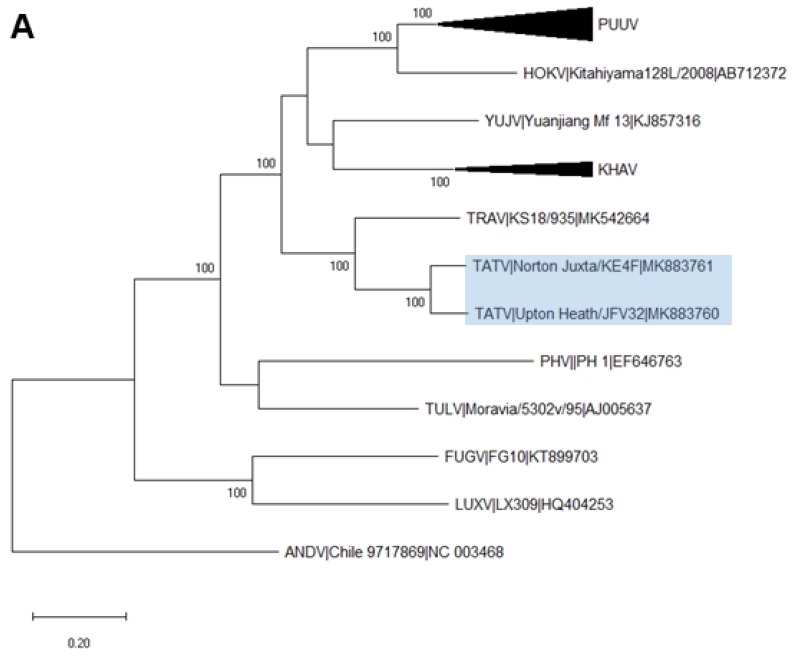
Phylogenetic relationship of Tatenale virus with other vole-associated orthohantavirus species. Representative complete coding sequences were retrieved for each segment; L (**A**), M (**B**) and S (**C**). Maximum likelihood trees were created with a GTR+G+I model, using MEGAX software. Branch lengths were drawn to a scale of nucleotide substitutions per site. L and S trees were based on full-length sequences, whilst the M segment tree was based on the available sequence for the partial Upton-Heath strain. Numbers above individual branches show bootstrap support after 1000 replicates. Tatenale virus strains are highlighted with a blue box. Black triangles represent a compressed species-specific subtree. Sequences are shown with the species name, strain name and the GenBank accession number. PUUV, Puumala virus; HOKV, Hokkaido virus; FUSV, Fusong virus; YUJV, Yuanjiang virus; KHAV, Khabarovsk virus; TOPV, Topografov virus; TATV, Tatenale virus; TRAV, Traemmersee virus; PHV, Prospect Hill virus; ILV, Isla Vista virus; TULV, Tula virus; ADLV, Adler virus; LUXV, Luxi virus; FUGV, Fugong virus; ANDV, Andes virus.

**Table 1 viruses-12-00454-t001:** The similarity of Norton-Juxta and Upton-Heath strains of Tatenale virus to the closest related strain of the most related species at nucleotide (amino acid) level. Similarities to the M segment of the Upton-Heath strain are based on the available partial sequence. * Indicates no complete sequence data available.

*Species (Accession Number)*	*S*	*M*	*L*
	**Norton-Juxta**		
*Traemersee*	82.7 (96.8)	79.8 (94.2)	81.5 (96.4)
*Khabarovsk*	79.2 (89.4)	76.4 (87.5)	77.9 (90.9)
*Yuanjiang*	79.2 (88.5)	75.3 (86.5)	77.7 (90.4)
*Fusong*	78.7 (88.2)	75.2 (85.9)	-*
*Puumala*	77.9 (87.8)	74.8 (84.7)	77.9 (88.1)
*Hokkaido*	78.3 (87.5)	75.5 (84.4)	76.8 (88.5)
	**Upton-Heath**		
*Traemersee*	83 (96.5)	80.8 (94.3)	81.5 (96.4)
*Khabarovsk*	79.9 (88.9)	77.1 (87.8)	78 (90.7)
*Yuanjiang*	78.9 (88.2)	75.7 (86.5)	77.7 (89.6)
*Fusong*	78.9 (88)	76 (86.2)	-*
*Puumala*	78.4 (87.8)	75.5 (84.6)	77.6 (87.5)
*Hokkaido*	79 (87.8)	75.7 (84.4)	76.7 (87.9)

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
