# Peer review of "Retrieval of the Complete Coding Sequence of the UK-Endemic Tatenale Orthohantavirus Reveals Extensive Strain Variation and Supports Its Classification as a Novel Species"

_viruses, 2020, doi:10.3390/v12040454_

Round 1
Reviewer 1 Report
The paper describes the complete nucleotide sequence of two strains of Tatenale/Traemmersee hantavirus which were found in field voles in England. These findings are of interest since they characterize a new hantavirus which probably forms a new virus species. Unfortunately, the writing of the ms. is far from being perfect.
Major concerns
- The scientific writing must be improved since the presentation is confusing and difficult to read. To give two examples only:
- The abstract is not understandable from itself. Line 27-29: is site 1 located in northern E. and site 2 in central E.?, line 29-32: unclear whether the strains were determined by different methods, line 32: what means third?, line 33: nothing new for northern E.?, line 33-36: novel species together with Traemmersee or not?
- The authors mix terms like site 1 and site 2, Norton-Juxta and Upton Heath, Central and Northern England without clear allocation.
- Make clear which name you want to propose for the tentative new species. Traemmersee was the first strain with complete sequence, however, partial sequences of Tatenale strains were published first. In the discussion about species demarcation you should follow the ICTV proposal no. 2018.010M available at https://talk.ictvonline.org/files/ictv_official_taxonomy_updates_since_the_8th_report/m/animal-dsrna-and-ssrna--viruses/8066
There, it is clearly stated that, based on the DEmARC analysis, species demarcation should be calculated and a species cutoff was defined:
Make an amino acid concatenated multiple alignment containing the full coding regions of the nucleocapsid protein (S segment) and glycoproteins (M segment);
• Calculate PED values using WAG amino acid substitution matrix (Tree-Puzzle, maximum likelihood parameter);
• A species of the genus Hantavirus is defined by a PED value greater than 0.1.
Therefore, I strongly recommend to perform this analysis and then modify this part accordingly.
Further points:
- The figures are too huge and overloaded with large numbers of unnecessary strain names. If you still want to include these strains in your phylogenetic analysis you can present them in condensed form by black triangles as also done in the paper of Jeske et al. 2019.
- Give an explanation why you used different methods to sequence the strains.
- Make clear in the Figure legend and Table that the M sequence of your “Norton-Juxta” strain is not “complete” as claimed in, e.g., line 172.
- Line 61: The yellow-necked mouse is not the only host of DOBV; different DOBV genotypes are associated with A. flavicollis, A. agrarius, and A. ponticus. This fact qualifies your statement in line 64.
- Check writing errors, as too much space between words (e.g., line 56, 71), missing space between numbers and measures (e.g., line 100-102), case sensitivity (e.g., line 104, 141), missing letters (e.g., line 109). Use numbers in line 137.
- Ref. 4: Delete [16], ref. 20: give the journal.
- Do not italicize the term Tatenale orthohantavirus since it is not a taxonomical term recognized by ICTV. I propose the term Tatenale virus or, alternatively, Traemmersee virus (see above).
Author Response
The scientific writing must be improved since the presentation is confusing and difficult to read. To give two examples only:
The abstract is not understandable from itself. Line 27-29: is site 1 located in northern E. and site 2 in central E.?, line 29-32: unclear whether the strains were determined by different methods, line 32: what means third?, line 33: nothing new for northern E.?, line 33-36: novel species together with Traemmersee or not?
The authors mix terms like site 1 and site 2, Norton-Juxta and Upton Heath, Central and Northern England without clear allocation.
We have updated the manuscript to improve the scientific writing and ensure that site and strain names are defined and used consistently.
Make clear which name you want to propose for the tentative new species. Traemmersee was the first strain with complete sequence, however, partial sequences of Tatenale strains were published first. In the discussion about species demarcation you should follow the ICTV proposal no. 2018.010M available at https://talk.ictvonline.org/files/ictv_official_taxonomy_updates_since_the_8th_report/m/animal-dsrna-and-ssrna--viruses/8066
There, it is clearly stated that, based on the DEmARC analysis, species demarcation should be calculated and a species cutoff was defined:
Make an amino acid concatenated multiple alignment containing the full coding regions of the nucleocapsid protein (S segment) and glycoproteins (M segment);
• Calculate PED values using WAG amino acid substitution matrix (Tree-Puzzle, maximum likelihood parameter);
• A species of the genus Hantavirus is defined by a PED value greater than 0.1.
Therefore, I strongly recommend to perform this analysis and then modify this part accordingly.
The pairwise evolutionary distance analysis recommended be performed, was used and the results of which are reported in section 3.2. We have added to the methods section of the manuscript describing this analysis. We have also proposed that the species be named Tatenale Orthohantavirus and given our reasons.
Further points:
The figures are too huge and overloaded with large numbers of unnecessary strain names. If you still want to include these strains in your phylogenetic analysis you can present them in condensed form by black triangles as also done in the paper of Jeske et al. 2019.
Trees have been modified to condense several species-specific sub-trees and increase their readability.
Give an explanation why you used different methods to sequence the strains.
We have added a brief sentence explaining that funding was not available to submit both strains for high-throughput sequencing.
Make clear in the Figure legend and Table that the M sequence of your “Norton-Juxta” strain is not “complete” as claimed in, e.g., line 172.
Legends have been updated to clarify that the Norton-Juxta M segment is incomplete.
Line 61: The yellow-necked mouse is not the only host of DOBV; different DOBV genotypes are associated with A. flavicollis, A. agrarius, and A. ponticus. This fact qualifies your statement in line 64.
We have updated the known reservoirs of DOBV to include A. agrarius. A. flavicollis and A. ponticus appear to be the considered the same species, and so the latter has been omitted.
Check writing errors, as too much space between words (e.g., line 56, 71), missing space between numbers and measures (e.g., line 100-102), case sensitivity (e.g., line 104, 141), missing letters (e.g., line 109). Use numbers in line 137.
Writing errors have been corrected
Ref. 4: Delete [16], ref. 20: give the journal.
References have been corrected
Do not italicize the term Tatenale orthohantavirus since it is not a taxonomical term recognized by ICTV. I propose the term Tatenale virus or, alternatively, Traemmersee virus (see above).
Italicised Tatenale orthohantavirus have been corrected.
Reviewer 2 Report
Has the almost complete CDS been finished? Should be included to full round out the story.
Many hantaviruses react with PUU/SNV nucleocapsid as it is Seoul virus also reacts with those same nucleocapsid. Can you proposed how you will look for specific reaction to TATV infection.
Was any serum collected to do staining with PUU nucleocapsid antigen? That would add more clinical relevance to the paper.
Author Response
Has the almost complete CDS been finished? Should be included to full round out the story.
We were unable to recover the complete M segment of the Norton-Juxta strain of TATV. We agree that it would help to complete the paper, however due to low viral titres in the primary material we utilised, this was not possible.
Many hantaviruses react with PUU/SNV nucleocapsid as it is Seoul virus also reacts with those same nucleocapsid. Can you proposed how you will look for specific reaction to TATV infection.
We are aware of the challenges of cross-reactivity in the serological diagnosis of orthohantavirus infection. Though not routine, focus reduction neutralization tests (FRNT) appear to be able to differentiate infection. We chose not to include significant reference to serology, as whilst an interesting area to further explore, it was beyond the aims and scope of this concise genome-focussed study.
Was any serum collected to do staining with PUU nucleocapsid antigen? That would add more clinical relevance to the paper.
Due to the nature of the collection method, in which animals were retrieved after being snap-trapped as part of pest control procedures, we were only able to retrieve solid tissue from the animals.
Reviewer 3 Report
Introduction
Line 71: New paragraph not needed before Furthermore.
Line 72: Was Tatenale virus identified directly from a vole and were there any human cases associated with it?
Line 80: Full stop at end of sentence.
Materials and Methods
2.1 Samples: More detail needed her. Coordinates of sites, or more specific locations required. Can you describe the sites? Were they urban, rural, farm, woodland etc? What type of traps did you use?
Line 86: What is the relevance of the reference to the previous paper? Are you using the same methods? It is not clear what this reference is referring to.
Line 87: site 2 needs capital S.
2.3 Rt-PCR Screening
Lines 97-98: Are these previously published primers, if so, a reference required, or did you design them?
Line 104: Second should have lowercase s.
Did you use the same amount of primer in the second assay and what was the concentration? You need to write out the primers as well as cite them.
2.4 High-Throughput Sequencing
Line 110: site 1 should have a capital S.
Line 117: were analyzed not was analyzed.
2.5 Retrieval of TATV CDS using PCR Primer-walking
Please define CDS and HTS (HTS can be defined in line 110).
Line 121: TATV Upton-Heath, what is this? Is this from Site 1, or Site 2 and from which rodent?
Line 123: Again what is TATV Norton Juxta, where and what is it from?
Results
You need to be consistent with capital letters - either site 1 or Site 1, sometimes it is with capitals and sometimes without. Please pick one and then change throughout and also for site 2.
Author Response
Introduction
Line 71: New paragraph not needed before Furthermore.
The new paragraph has been removed.
Line 72: Was Tatenale virus identified directly from a vole and were there any human cases associated with it?
TATV has only been identified in Field voles, the manuscript has been updated to clarify this.
Line 80: Full stop at end of sentence.
Materials and Methods
2.1 Samples: More detail needed her. Coordinates of sites, or more specific locations required. Can you describe the sites? Were they urban, rural, farm, woodland etc? What type of traps did you use?
We have added the coordinates of the sites into the manuscript, along with a very brief description. Rodents were killed with snap-traps, as part of routine pest-control measures at both sites, we don’t have any further information regarding the traps than this unfortunately.
Line 86: What is the relevance of the reference to the previous paper? Are you using the same methods? It is not clear what this reference is referring to.
Corrected (this was indeed erroneously included in this context)
Line 87: site 2 needs capital S.
Corrected
2.3 Rt-PCR Screening
Lines 97-98: Are these previously published primers, if so, a reference required, or did you design them?
Primers were designed in-house. The manuscript has been updated to include this.
Line 104: Second should have lowercase s.
Corrected
Did you use the same amount of primer in the second assay and what was the concentration? You need to write out the primers as well as cite them.
Primer volumes and concentrations were the same in both assays. The manuscript has been updated to clarify this, as well as adding the primer sequences.
2.4 High-Throughput Sequencing
Line 110: site 1 should have a capital S.
Corrected
Line 117: were analyzed not was analyzed.
Corrected
2.5 Retrieval of TATV CDS using PCR Primer-walking
Please define CDS and HTS (HTS can be defined in line 110).
Definitions have been added for both terms.
Line 121: TATV Upton-Heath, what is this? Is this from Site 1, or Site 2 and from which rodent?
TATV Upton-Heath was the strain recovered from a field vole captured at site 2 (Cheshire). The manuscript has been reworded to remove this ambiguity.
Line 123: Again what is TATV Norton Juxta, where and what is it from?
TATV Norton-Juxta was recovered from Site 1 (Leicestershire). As above, the manuscript has been reworded for clarification.
You need to be consistent with capital letters - either site 1 or Site 1, sometimes it is with capitals and sometimes without. Please pick one and then change throughout and also for site 2.
Site names have been updated for consistency
Round 2
Reviewer 1 Report
I found items which were not corrected during revision:
- line 3: omit Italics
- line 33: What does “third lineage” mean? Your two strains together with the partially sequenced Tatenale strains published before? Or do you mean your strains together with Traemersee? Explain here and in the text.
- In any case, you should compare the available sequences of the older Tatenale strains and your stains.
Author Response
1. line 3: omit Italics
Italics have been removed
2. line 33: What does “third lineage” mean? Your two strains together with the partially sequenced Tatenale strains published before? Or do you mean your strains together with Traemersee? Explain here and in the text.
The ‘third lineage’ refers to the Norton-Juxta strain of Tatenale. The Upton-Heath strain is highly similar to the previously reported B41 strain, whilst the kld (Kielder) and Norton-Juxta strains cluster separately on the phylogenetic trees based on the partial sequences available.
3. In any case, you should compare the available sequences of the older Tatenale strains and your stains.
Comparison of the S segment of B41 with Norton-Juxta and Upton-Heath has been added. L Segment comparison is shown in lines 150-153. A Phylogenetic tree including the available partial sequences of the TATV B41 and Kielder strains have been included as a supplementary figure.
Reviewer 3 Report
Line 68: Define United Kingdom (UK) at first use and then use UK thereafter – check whole manuscript please.
Line 75: Include full genus name of field vole, Microtus agrestis not just M. agrestis.
Line 82: Full stop at end of sentence.
Line 91: Please include the following: “Rodents were captured as part of routine pest-management at both sites.”
Line 126: “Complete coding sequences” should be “complete coding sequence”.
Line 152: Insert a space between 1 and virus.
Line 163: Complete does not need a capital C.
Line 168: were not was.
Line 179: Amino acid does not need capital A.
Line 194: species-specific not species, specific.
Line 203: do not capitalise amino acid.
Table 1: Upton-Heath not Upton Heath. You have hyphenated this up until this table so be consistent. Also in the title/legend for Table one, you need to hyphenate the site names and on line 204, capitalise heath. Full stop at the end of sentence on line 205.
Line 216: This is the first time you have abbreviated amino acid, so please define properly.
Line 225: Should read as “PUUV- or SNV-like viruses”.
Line 227: Change Pounder et al to “A previous study” and move [20] to the end of the sentence.
Author Response
Line 68: Define United Kingdom (UK) at first use and then use UK thereafter – check whole manuscript please.
UK has been defined
Line 75: Include full genus name of field vole, Microtus agrestis not just M. agrestis.
Corrected
Line 82: Full stop at end of sentence.
Corrected
Line 91: Please include the following: “Rodents were captured as part of routine pest-management at both sites.”
Sentence has been added
Line 126: “Complete coding sequences” should be “complete coding sequence”.
Corrected
Line 152: Insert a space between 1 and virus.
Corrected
Line 163: Complete does not need a capital C.
Corrected
Line 168: were not was.
Corrected
Line 179: Amino acid does not need capital A.
Corrected
Line 194: species-specific not species, specific.
Corrected
Line 203: do not capitalise amino acid.
Corrected
Table 1: Upton-Heath not Upton Heath. You have hyphenated this up until this table so be consistent. Also in the title/legend for Table one, you need to hyphenate the site names and on line 204, capitalise heath. Full stop at the end of sentence on line 205.
Corrected
Line 216: This is the first time you have abbreviated amino acid, so please define properly.
Corrected
Line 225: Should read as “PUUV- or SNV-like viruses”.
Corrected
Line 227: Change Pounder et al to “A previous study” and move [20] to the end of the sentence.
Corrected